Quantifying the impact of non-stationarity in reinforcement learning-based traffic signal control

http://orcid.org/0000-0001-5465-4390 Alegre Lucas N. 1 lnalegre@inf.ufrgs.br
http://orcid.org/0000-0002-2803-9607 Bazzan Ana L.C. 1
da Silva Bruno C. 2
1 Institute of Informatics, Universidade Federal do Rio Grande do Sul , Porto Alegre, Rio Grande do Sul , Brazil
2 CICS, University of Massachusetts , Amherst, Massachusetts , United States of America
Gao Zhiwei
Electronic publication date: 2021 May 27
Publication date: 2021
Volume: 7
Electronic Location ID: e575
Received 2020 Nov 4; Accepted 2021 May 11
Copyright: © 2021 Alegre et al.
Copyright year: 2021
Copyright holder: Alegre et al.
License: This is an open access article distributed under the terms of the Creative Commons Attribution License, which permits unrestricted use, distribution, reproduction and adaptation in any medium and for any purpose provided that it is properly attributed. For attribution, the original author(s), title, publication source (PeerJ Computer Science) and either DOI or URL of the article must be cited.
License URL: https://creativecommons.org/licenses/by/4.0/

Keywords: Reinforcement learning, Traffic signal control, Non-stationarity, Multiagent systems

Funding: CNPq 140500/2021-9 CNPq 307215/2017-2 Lucas N. Alegre was supported by CNPq under grant no. 140500/2021-9. Ana Bazzan was supported by CNPq under grant no. 307215/2017-2. This study was financed in part by the Coordenação de Aperfeiçoamento de Pessoal de Nível Superior - Brasil (CAPES) - Finance Code 001. There was no additional external funding received for this study. The funders had no role in study design, data collection and analysis, decision to publish, or preparation of the manuscript.

==============================
In reinforcement learning (RL), dealing with non-stationarity is a challenging issue. However, some domains such as traffic optimization are inherently non-stationary. Causes for and effects of this are manifold. In particular, when dealing with traffic signal controls, addressing non-stationarity is key since traffic conditions change over time and as a function of traffic control decisions taken in other parts of a network. In this paper we analyze the effects that different sources of non-stationarity have in a network of traffic signals, in which each signal is modeled as a learning agent. More precisely, we study both the effects of changing the context in which an agent learns (e.g., a change in flow rates experienced by it), as well as the effects of reducing agent observability of the true environment state. Partial observability may cause distinct states (in which distinct actions are optimal) to be seen as the same by the traffic signal agents. This, in turn, may lead to sub-optimal performance. We show that the lack of suitable sensors to provide a representative observation of the real state seems to affect the performance more drastically than the changes to the underlying traffic patterns.

Introduction

Controlling traffic signals is one way of dealing with the increasing volume of vehicles that use the existing urban network infrastructure. Reinforcement learning (RL) adds up to this effort by allowing decentralization (traffic signals—modeled as agents—can independently learn the best actions to take in each current state) as well as on-the-fly adaptation to traffic flow changes. It is noteworthy that this can be done in a model-free way (with no prior domain information) via RL techniques. RL is based on an agent computing a policy mapping states to actions without requiring an explicit environment model. This is important in traffic domains because such a model may be very complex, as it involves modeling traffic state transitions determined not only by the actions of multiple agents, but also by changes inherent to the environment—such as time-dependent changes to the flow of vehicles.

One of the major difficulties in applying reinforcement learning (RL) in traffic control problems is the fact that the environments may change in unpredictable ways. The agents may have to operate in different contexts—which we define here as the true underlying traffic patterns affecting an agent; importantly, the agents do not know the true context of their environment, e.g., since they do not have full observability of the traffic network. Examples of partially observable variables that result in different contexts include different traffic patterns during the hours of the day, traffic accidents, road maintenance, weather, and other hazards. We refer to changes in the environment’s dynamics as non-stationarity.

In terms of contributions, we introduce a way to model different contexts that arise in urban traffic due to time-varying characteristics. We then analyze different sources of non-stationarity—when applying RL to traffic signal control—and quantify the impact that each one has on the learning process. More precisely, we study the impact in learning performance resulting from (1) explicit changes in traffic patterns introduced by different vehicle flow rates; and (2) reduced state observability resulting from imprecision or unavailability of readings from sensors at traffic intersections. The latter problem may cause distinct states (in which distinct actions are optimal) to be seen as the same by the traffic signal agents. This not only leads to sub-optimal performance but may introduce drastic drops in performance when the environment’s context changes. We evaluate the performance of deploying RL in a non-stationary multiagent scenario, where each traffic signal uses Q-learning—a model-free RL algorithm—to learn efficient control policies. The traffic environment is simulated using the open-source microscopic traffic simulator SUMO (Simulation of Urban MObility) (Lopez et al., 2018) and models the dynamics of a 4 × 4 grid traffic network with 16 traffic signal agents, where each agent has access only to local observations of its controlled intersection. We empirically demonstrate that the aforementioned causes of non-stationarity can negatively affect the performance of the learning agents. We also demonstrate that the lack of suitable sensors to provide a representative observation of the true underlying traffic state seems to affect learning performance more drastically than changes to the underlying traffic patterns.

The rest of this paper is organized as follows. The next section briefly introduces relevant RL concepts. Then, our model is introduced in “Methods”, and the corresponding experiments in “Experiments and Results”. Finally, we discuss related work in “Related Work” and then present concluding remarks.

Background

Reinforcement learning

In reinforcement learning (Sutton & Barto, 1998), an agent learns how to behave by interacting with an environment, from which it receives a reward signal after each action. The agent uses this feedback to iteratively learn an optimal control policy π *—a function that specifies the most appropriate action to take in each state. We can model RL problems as Markov decision processes (MDPs). These are described by a set of states S, a set of actions A, a reward function R(s,a,s′)→R and a probabilistic state transition function T(s,a,s′)→[0,1]. An experience tuple〈s, a, s′, r〉denotes the fact that the agent was in state s, performed action a and ended up in s′ with reward r. Let t denote the tth step in the policy π. In an infinite horizon MDP, the cumulative reward in the future under policy π is defined by the action-value function (or Q-function) Qπ(s,a), as in Eq. (1), where γ ∈ [0, 1] is the discount factor for future rewards.

(1) Qπ(s,a)=E[∑τ=0∞γτrt+τ|st=s,at=a,π]

If the agent knows the optimal Q-values Q*(s,a) for all state-actions pairs, then the optimal control policy π* can be easily obtained; since the agent’s objective is to maximize the cumulative reward, the optimal control policy is:

(2) π∗(s)=argmaxaQ∗(s,a)∀s∈S,a∈A

Reinforcement learning methods can be divided into two categories: model-free and model-based. Model-based methods assume that the transition function T and the reward function R are available, or instead try to learn them. Model-free methods, on the other hand, do not require that the agent have access to information about how the environment works. Instead, they learn an action-value function based only on samples obtained by interacting with the environment.

The RL algorithm used in this paper is Q-learning (QL), a model-free off-policy algorithm that estimates the Q-values in the form of a Q-table. After an experience〈s, a, s′, r〉, the corresponding Q(s,a) value is updated through Eq. (3), where α ∈ [0, 1] is the learning rate.

(3) Q(s,a):=Q(s,a)+α(r+γmaxa⁡Q(s′,a)−Q(s,a))

Importantly, in the tabular case with online learning, which we tackle in our work, Q-learning is known to converge to optimal policies given mild assumptions about exploration whenever deployed on stationary MDPs (Watkins, 1989; Tsitsiklis, 1994).

In order to balance exploitation and exploration when agents select actions, we use in this paper the ε-greedy mechanism. This way, agents randomly explore with probability ε and choose the action with the best expected reward so far with probability 1 − ε.

Non-stationarity in RL

In RL, dealing with non-stationarity is a challenging issue (Hernandez-Leal et al., 2017). Among the main causes of non-stationarity are changes in the state transition function T(s, a, s′) or in the reward function R(s, a, s′), partial observability of the true environment state (discussed in “Partial Observability”) and non-observability of the actions taken by other agents.

In an MDP, the probabilistic state transition function T is assumed not to change. However, this is not realistic in many real world problems. In non-stationary environments, the state transition function T and/or the reward function R can change at arbitrary time steps. In traffic domains, for instance, an action in a given state may have different results depending on the current context—i.e., on the way the network state changes in reaction to the actions of the agents. If agents do not explicitly deal with context changes, they may have to readapt their policies. Hence, they may undergo a constant process of forgetting and relearning control strategies. Though this readaptation is possible, it might cause the agent to operate in a sub-optimal manner for extended periods of time.

Importantly, no convergence guarantees exist in the non-stationary case, and so one needs to design ways to keep the agents from being heavily affected by changes to the environment’s dynamics (Padakandla, 2020). Motivated by this challenge, one of the goals of our work is to quantify the impact that different sources of non-stationarity have on the agents’ learning process. Ideally, one should aim to shape the learning problem into one that is as stationary as possible, so that convergence guarantees may be given. Recent work in the RL literature has investigated methods for dealing with non-stationary environments by explicitly modeling a set of contexts and their associated local policies (Alegre, Bazzan & Da Silva, 2021; Padakandla, 2020). These methods are orthogonal to the idea studied in our paper: by augmenting state definitions we can reduce partial observability and thus minimize the effect of non-stationarity on the learning process and on convergence.

Partial observability

Traffic control problems might be modeled as Dec-POMDPs (Bernstein, Zilberstein & Immerman, 2000)—a particular type of decentralized multiagent MDP where agents have only partial observability of their true states. A Dec-POMDP introduces to an MDP a set of agents I, for each agent i ∈ I a set of actions Ai, with A = XiAi the set of joint actions, a set of observations Ωi, with Ω = XiΩi the set of joint observations, and observation probabilities O(o|s,a), the probability of agents seeing observations o, given the state is s and agents take actions a. As specific methods to solve Dec-POMDPs do not scale with the number of agents (Bernstein et al., 2002), it is usual to tackle them using techniques conceived to deal with the fully-observable case. Though this allows for better scalability, it introduces non-stationarity as the agents cannot completely observe their environment nor the actions of other agents.

In traffic signal control, partial observability can appear due to lack of suitable sensors to provide a representative observation of the traffic intersection. Additionally, even when multiple sensors are available, partial observability may occur due to inaccurate (with low resolution) measures.

Methods

As mentioned earlier, the main goal of this paper is to investigate the different causes of non-stationarity that might affect performance in a scenario where traffic signal agents learn how to improve traffic flow under various forms of non-stationarity. To study this problem, we introduce a framework for modeling urban traffic under time-varying dynamics. In particular, we first introduce a baseline urban traffic model based on MDPs. This is done by formalizing—following similar existing works—the relevant elements of the MDP: its state space, action set, and reward function.

Then, we show how to extend this baseline model to allow for dynamic changes to its transition function so as to encode the existence of different contexts. Here, contexts correspond to different traffic patterns that may change over time according to causes that might not be directly observable by the agent. We also discuss different design decisions regarding the possible ways in which the states of the traffic system are defined; many of these are aligned with the modeling choices typically done in the literature, as for instance (Mannion, Duggan & Howley, 2016; Genders & Razavi, 2018). Discussing the different possible definitions of states is relevant since these are typically specified in a way that directly incorporates sensor information. Given the amount and quality of sensor information, however, different state definitions arise that—depending on sensor resolution and partial observability of the environment and/or of other agents—result in different amounts of non-stationarity.

Furthermore, in what follows we describe the multiagent training scheme used (in “Multiagent Independent Q-learning”) by each traffic signal agent in order to optimize its policy under non-stationary settings. We also describe how traffic patterns—the contexts in which our agents may need to operate—are modeled mathematically in “Contexts”. We discuss the methodology that is used to analyze and quantify the effects of non-stationarity in the traffic problem in “Experiments and Results”.

Finally, we emphasize here that the proposed methods and analyzes that will be conducted in this paper—aimed at evaluating the impact of different sources of non-stationary—are the main contributions of our work. Most existing works (e.g., those discussed in “Related Work”) do not address or directly investigate at length the implications of varying traffic flow rates as sources of non-stationarity in RL.

State formulation

In the problems or scenarios we deal with, the definition of state space strongly influences the agents’ behavior and performance. Each traffic signal agent controls one intersection, and at each time step t it observes a vector st that partially represents the true state of the controlled intersection.

A state, in our problem, could be defined as a vector s∈R(2+2|P|), as in Eq. (4), where P is the set of all green traffic phases1 , ρ ∈ P denotes the current green phase, δ ∈ [0, maxGreenTime] is the elapsed time of the current phase, densityi ∈ [0, 1] is defined as the number of vehicles divided by the vehicle capacity of the incoming movements of phase i and queuei ∈ [0, 1] is defined as the number of queued vehicles (we consider as queued a vehicle with speed under 0.1 m/s) divided by the vehicle capacity of the incoming movements of phase i.

(4) s=[ρ,δ,density1,queue1,...,density|P|,queue|P|]

Note that this state definition might not be feasibly implementable in real-life settings due to cost issues arising from the fact that many physical sensors would have to be paid for and deployed. We introduce, for this reason, an alternative definition of state which has reduced scope of observation. More precisely, this alternative state definition removes density attributes from Eq. (4), resulting in the partially-observable state vector s∈R(2+|P|) in Eq. (5). The absence of these state attributes is analogous to the lack of availability of real-life traffic sensors capable of detecting approaching vehicles along the extension of a given street (i.e., the density of vehicles along that street). This implies that, without the density attributes, the observed state can not inform the agent whether (or how fast) the links are being filled with new incoming vehicles, which may lead to a situation with large queue lengths in the next time steps.

(5) s=[ρ,δ,queue1,...,queue|P|]

Note also that the above definition results in continuous states. Q-learning, however, traditionally works with discrete state spaces. Therefore, states need to be discretized after being computed. Both density and queue attributes are discretized in ten levels/bins equally distributed. We point out that a low level of discretization is also a form of partial-observability, as it may cause distinct states to be perceived as the same state. Furthermore, in this paper we assume—as commonly done in the literature—that one simulation time step corresponds to five seconds of real-life traffic dynamics. This helps encode the fact that traffic signals typically do not change actions every second; this modeling decision implies that actions (in particular, changes to the current phase of a traffic light) are taken in intervals of five seconds.

Actions

In an MDP, at each time step t each agent chooses an action at ∈ A. The number of actions, in our setting, is equal to the number of phases, where a phase allows green signal to a specific traffic direction; thus, |A| = |P|. In the case where the traffic network is a grid (typically encountered in the literature (El-Tantawy, Abdulhai & Abdelgawad, 2013; Mannion, Duggan & Howley, 2016; Chu et al., 2019)), we consider two actions: an agent can either keep green time to the current phase or allow green time to another phase; we call these actions keep and change, respectively. There are two restrictions in the action selection: an agent can take the action change only if δ ≥ 10 s (minGreenTime) and the action keep only if δ < 50 s (maxGreenTime). Additionally, change actions impose a yellow phase with a fixed duration of 2 s. These restrictions are in place to, e.g., model the fact that in real life, a traffic controller needs to commit to a decision for a minimum amount of time to allow stopped cars to accelerate and move to their intended destinations.

Reward function

The rewards assigned to traffic signal agents in our model are defined as the change in cumulative vehicle waiting time between successive actions. After the execution of an action at, the agent receives a reward rt∈R as given by Eq. (6):

(6) rt=Wt−Wt+1

where Wt and Wt + 1 represent the cumulative waiting time at the intersection before and after executing the action at, following Eq. (7):

(7) Wt=∑v∈Vtwv,t

where Vt is the set of vehicles on roads arriving at an intersection at time step t, and wv, t is the total waiting time of vehicle v since it entered one of the roads arriving at the intersection until time step t. A vehicle is considered to be waiting if its speed is below 0.1 m/s. Note that, according to this definition, the larger the decrease in cumulative waiting time, the larger the reward. Consequently, by maximizing rewards, agents reduce the waiting time at the intersections, thereby improving the local traffic flow.

Multiagent independent Q-learning

We tackle the non-stationarity in our scenario by using Q-learning in a multiagent independent training scheme (Tan, 1993), where each traffic signal is a QL agent with its own Q-table, local observations, actions and rewards. This approach allows each agent to learn an individual policy, applicable given the local observations that it makes; policies may vary between agents as each one updates its Q-table using only its own experience tuples. Besides allowing for different behaviors between agents, this approach also avoids the curse of dimensionality that a centralized training scheme would introduce. However, there is one main drawback of an independent training scheme: as agents are learning and adjusting their policies, changes to their policies cause the environment dynamics to change, thereby resulting in non-stationary. This means that original convergence properties for single-agent algorithms no longer hold due to the fact that the best policy for an agent changes as other agents’ policies change (Busoniu, Babuska & De Schutter, 2008).

Contexts

In order to model one of the causes for non-stationary in the environment, we use the concept of traffic contexts, similarly to Da Silva et al. (2006). We define contexts as traffic patterns composed of different vehicle flow distributions over the Origin-Destination (OD) pairs of the network. The origin node of an OD pair indicates where a vehicle is inserted in the simulation. The destination node is the node in which the vehicle ends its trip, and hence is removed from the simulation upon its arrival. A context, then, is defined by associating with each OD pair a number of vehicles that are inserted (per second) in its origin node. Non-stationarity then emerges since the current context changes during the simulation in the form of recurrent events on the traffic environment. Importantly, although each context corresponds to a stationary traffic pattern, the environment becomes non-stationary w.r.t. the agents because the underlying context changes unpredictably, and the agents cannot perceive an indicator of the current context.

Changing the context during a simulation causes the sensors measures to vary differently in time. Events such as traffic accidents and hush hours, for example, cause the flow of vehicles to increase in a particular direction, thus making the queues on the lanes of this direction to increase faster. In the usual case, where agents do not have access to all information about the environment state, this can affect the state transition T and the reward R functions of the MDP directly. Consequently, when the state transition probabilities and the rewards agents are observing change, the Q-values of the state-action pairs also change. Therefore, traffic signal agents will most likely need to undergo a readaptation phase to correctly update their policies, resulting in periods of catastrophic drops in performance.

Experiments and results

Our main goal with the following experiments is to quantify the impact of different causes of non-stationarity in the learning process of an RL agent in traffic signal control. Explicit changes in context (e.g., vehicle flow rate changes in one or more directions) are one of these causes and are present in all of the following experiments. This section first describes details of the scenario being simulated as well as the traffic contexts, followed by a definition of the performance metrics used as well as the different experiments that were performed.

We first conduct an experiment where traffic signals use a fixed control policy—a common strategy in case the infrastructure lacks sensors and/or actuators. The results of this experiment are discussed in “Traffic Signal Control under Fixed Policies” and are used to emphasize the problem of lacking a policy that can adapt to different contexts; it also serves as a baseline for later comparisons. Afterwards, in “Effects of Disabling Learning and Exploration” we explore the setting where agents employ a given policy in a context/traffic pattern that has not yet been observed during the training phase. In “Effects of Reduced State Observability” we analyze (1) the impact of context changes when agents continue to explore and update their Q-tables throughout the simulation; and (2) the impact of having non-stationarity introduced both by context changes and by the use of the two different state definitions presented in “State Formulation”. Then, in “Effects of Different Levels of State Discretization” we address the relation between non-stationarity and partial observations resulting from the use of imprecise sensors, simulated by poor discretization of the observation space. Lastly, in “Discussion” we discuss what are the main findings and implications of the results observed.

Scenario

We used the open-source microscopic traffic simulator SUMO to model and simulate the traffic scenario and its dynamics, and SUMO-RL (Alegre, 2019) to instantiate the simulation as a reinforcement learning environment with all the components of an MDP. The traffic network is a 4 × 4 grid network with traffic signals present in all 16 intersections (Fig. 1). All links have 150 m, two lanes and are one-way. Vertical links follow N-S traffic directions and horizontal links follow W-E directions. There are eight OD pairs: 4 in the W-E traffic direction (A2F2, A3F3, A4F4, and A5F5), and 4 in the N-S direction (B1B6, C1C6, D1D6, E1E6).

Figure 1 4 × 4 grid network.

(A) Network topology. (B) Network in SUMO.

In order to demonstrate the impact of context changes on traffic signals (and hence, on the traffic), we defined two different traffic contexts with different vehicle flow rates. Both contexts insert the same amount of vehicles per second in the network, but do so by using a different distribution of those vehicles over the possible OD pairs. In particular:Context 1 (NS = WE): insertion rate of 1 vehicle every 3 s in all eight OD pairs.

Context 2 (NS<WE): insertion rate of 1 vehicle every 6 s in the N-S direction OD pairs and one vehicle every 2 s in the W-E direction OD pairs.

It is expected that a policy in which the two green traffic phases are equally distributed would have a satisfactory performance in Context 1, but not in Context 2. In the following experiments, we shift between Context 1 and Context 2 every 20,000 time steps, starting the simulation with Context 1. This means that the insertion rates change every 20,000 time steps, following the aforementioned contexts.

Metrics

To measure the performance of traffic signal agents, we used as metric the summation of the cumulative vehicle waiting time on all intersections, as in Eq. (7). Intuitively, this quantifies for how long vehicles are delayed by having to reduce their velocity below 0.1 m/s due to long waiting queues and to the inadequate use of red signal phases. This metric is also a good indication of the agents performance, since it is strongly related to the rewards assigned to each agent, defined in Eq. (6). Therefore, as the agents improve their local policies to minimize the change in cumulative vehicle waiting time, it is expected that the global waiting time of the traffic environment also decreases.

At the time steps in which phase changes occur, natural oscillations in the queue sizes occur since many vehicles are stopping and many are accelerating. Therefore, all plots shown here depict moving averages of the previously-discussed metric within a time window of 15 s. The plots related to Q-learning are averaged over 30 runs, where the shadowed area shows the standard deviation. Additionally, we omit the time steps of the beginning of the simulation (since the network then is not yet fully populated with vehicles) as well as the last time steps (since then vehicles are no longer being inserted).

Traffic signal control under fixed policies

We first demonstrate the performance of a fixed policy designed by following the High Capacity Manual (National Research Council, 2000), which is popularly used for such task. The fixed policy assigns to each phase a green time of 35 s and a yellow time of 2 s. As mentioned, our goal by defining this policy is to construct a baseline used to quantify the impact of a context change on the performance of traffic signals in two situations: one where traffic signals follow a fixed policy and one where traffic signals adapt and learn a new policy using QL algorithm. This section analyzes the former case. Figure 2 shows that the fixed policy, as expected, loses performance when the context is changed. When the traffic flow is set to Context 2 at time step 20,000, a larger amount of vehicles are driving in the W-E direction and thus producing larger waiting queues. In order to obtain a good performance using fixed policies, it would be necessary to define a policy for each context and to know in advance the exact moment when context changes will occur. Moreover, there may be an arbitrarily large number of such contexts, and the agent, in general, has no way of knowing in advance how many exist. Prior knowledge of these quantities is not typically available since non-recurring events that may affect the environment dynamics, such as traffic accidents, cannot be predicted. Hence, traffic signal control by fixed policies is inadequate in scenarios where traffic flow dynamics may change (slowly or abruptly) over time.

Figure 2 Total waiting time of vehicles in the simulation: fixed policy traffic signals, context change at time step 20,000.

Effects of disabling learning and exploration

We now describe the case in which agents stop, at some point in time, to learn from their actions and simply follow the policy learned before a given context change. The objective here is to simulate a situation where a traffic signal agent employs a previously-learned policy to a context/traffic pattern that has not yet been observed in its training phase. We achieve this by setting both α (learning rate) and ε (exploration rate) to 0 when there is a change in context. By observing Eq. (3), we see that the Q-values no longer have their values changed if α = 0. By setting ε = 0, we also ensure that the agents will not explore and that they will only choose the actions with the higher estimated Q-value given the dynamics of the last observed context. By analyzing performance in this setting, we can quantify the negative effect of agents that act solely by following the policy learned from the previous contexts.

During the training phase (until time step 20,000), we use a learning rate of α = 0.1 and discount factor γ = 0.99. The exploration rate starts at ε = 1 and decays by a factor of 0.9985 every time the agent chooses an action. These definitions ensure that the agents are mostly exploring at the beginning, while by the time step 10,000 ε is below 0.05, thereby resulting in agents that continue to purely exploit a currently-learned policy even after a context change; i.e., agents that do not adapt to context changes.

In Fig. 3 we observe that the total waiting time of vehicles rapidly increases after the context change (time step 20,000). This change in the environment dynamics causes the policy learned in Context 1 to no longer be efficient, since Context 2 introduces a flow pattern that the traffic signals have not yet observed. Consequently, the traffic signal agents do not know what are the best actions to take when in those states. Note, however, that some actions (e.g., changing the phase when there is congestion in one of the directions) are still capable of improving performance, since they are reasonable decisions under both contexts. This explains why performance drops considerably when the context changes and why the waiting time keeps oscillating afterwards.

Figure 3 Total waiting time of vehicles.

Q-learning traffic signals. Context change α and ε set to 0 at timestep 20,000.

Effects of reduced state observability

In this experiment, we compare the effects of context changes under the two different state definitions presented in “State Formulation”. The state definition in Eq. (4) represents a more unrealistic scenario in which expensive real-traffic sensors are available at the intersections. In contrast, in the partial state definition in Eq. (5) each traffic signal has information only about how many vehicles are stopped at its corresponding intersection (queue), but cannot relate this information to the number of vehicles currently approaching its waiting queue, as vehicles in movement are monitored only on density attributes.

Differently from the previous experiment, agents now continue to explore and update their Q-tables throughout the simulation. The ε parameter is set to a fixed value of 0.05; this way, the agents mostly exploit but still have a small chance of exploring other actions in order to adapt to changes in the environment. By not changing ε we ensure that performance variations are not caused by an exploration strategy. The values of the QL parameters (α and γ) are kept as in the previous experiment.

The results of this experiment are shown in Fig. 4. By analyzing the initial steps in the simulation, we note that agents using the reduced state definition learn significantly faster than those with the state definition that incorporates both queue and density attributes. This is because there are fewer states to explore, and so it takes fewer steps for the policy to converge. However, given this limited observation capability, agents converge to a policy resulting in higher waiting times when compared to that resulting from agents with more extensive state observability. This shows that the density attributes are fundamental to better characterize the true state of a traffic intersection. Also note that around time 10,000, the performance of both state definitions (around 500 s of total waiting time) are better than that achieved under the fixed policy program (around 2,200 s of total waiting time), depicted in Fig. 2.

Figure 4 Total waiting time of vehicles.

Q-learning agents with two state representations: queue and queue + density. Context changes at times 20,000, 40,000 and 60,000.

In the first context change, at time 20,000, the total waiting time of both state definitions increases considerably. This is expected as it is the first time agents have to operate in Context 2. Agents operating under the original state definition recovered from this context change rapidly and achieved the same performance obtained in Context 1. However, with the partial state definition (i.e., only queue attributes), it is more challenging for agents to behave properly when operating under Context 2, which depicts an unbalanced traffic flow arriving at the intersection.

Finally, we can observe how (at time step 60,000) the non-stationarity introduced by context changes relates to the limited partial state definition. While traffic signal agents observing both queue and density do not show any oscillations in the waiting time of their controlled intersections, agents observing only queue have a significant performance drop. Despite having already experienced Context 2, they had to relearn their policies since the past Q-values were overwritten by the learning mechanism to adapt to the changing past dynamics. The dynamics of both contexts are, however, well-captured in the original state definition, as the combination of the density and queue attributes provides enough information about the dynamics of traffic arrivals at the intersection. This observation emphasizes the importance of more extensive state observability to avoid the negative impacts of non-stationarity in RL agents.

Effects of different levels of state discretization

Besides the unavailability of appropriate sensors (which results in incomplete description of states) another possible cause of non-stationarity is poor precision and low range of observations. As an example, consider imprecision in the measurement of the number of vehicles waiting at an intersection; this may cause distinct states—in which distinct actions are optimal—to be perceived as the same state. This not only leads to sub-optimal performance, but also introduces drastic performance drops when the context change. We simulate this effect by lowering the number of discretization levels of the attribute queue in cases where the density attribute is not available.

In Fig. 5 we depict how the discretization level of the attribute queue affects performance when a context change occurs. The red line corresponds to the performance when queue is discretized into 10 equally-distributed levels/bins (see “State Formulation”). The dark blue line corresponds to performance under a reduced discretization level of 4 bins. Note how after a context change (at time steps 20,000, 40,000 and 60,000) we can observe how the use of reduced discretization levels causes a significant drop in performance. At time 40,000, for instance, the total waiting time increases up to three times when operating under the lower discretization level.

Figure 5 Total waiting time of vehicles.

Q-learning traffic signals with different levels of discretization for the attribute queue. Context changes at time steps 20,000, 40,000 and 60,000.

Intuitively, an agent with imprecise observation of its true state has reduced capability to perceive changes in the transition function. Consequently, when traffic flow rates change at an intersection, agents with imprecise observations require a larger number of actions to readapt, thereby dramatically increasing queues.

Discussion

Many RL algorithms have been proposed to tackle non-stationary problems (Choi, Yeung & Zhang, 2000; Doya et al., 2002; Da Silva et al., 2006). Specifically, these works assume that the environment is non-stationary (without studying or analyzing the specific causes of non-stationary) and then propose computational mechanisms to efficiently learn under that setting. In this paper, we deal with a complementary problem, which is to quantify the effects of different causes of non-stationarity in the learning performance. We also assume that non-stationarity exists, but we explicitly model many of the possible underline reasons why its effects may take place. We study this complementary problem because it is our understanding that by explicitly quantifying the different reasons for non-stationary effects, it may be possible to make better-informed decisions about which specific algorithm to use, or to decide, for instance, if efforts should be better spent by designing a more complete set of features instead of by designing more sophisticated learning algorithms.

In this paper, we studied these possible causes specifically when they affect urban traffic environments. The results of our experiments indicate that non-stationarity in the form of changes to vehicle flow rates significantly impact both traffic signal controllers following fixed policies and policies learned from standard RL methods that do not model different contexts. However, this impact (that results in rapid changes in the total number of vehicles waiting at the intersections) has different levels of impact on agents depending on the different levels of observability available to those agents. While agents with the original state definition (queue and density attributes) only present performance drops in the first time they operate in a new context, agents with reduced observation (only queue attributes) may always have to relearn the readapted Q-values. The original state definition, however, is not very realistic in the real world, as sensors capable of providing both attributes for large traffic roads are very expensive. Finally, in cases where agents observe only the queues attributes, we demonstrated that imprecise measures (e.g. low number of discretization bins) potencializes the impact of context changes. Hence, in order to design a robust RL traffic signal controller, it is critical to take into account which are the most adequate sensors and how they contribute to provide a more extensive observation of the true environment state.

We observed that the non-stationarity introduced by the actions of other concurrently-learning agents in a competitive environment seemed to be a minor obstacle to acquiring effective traffic signals policies. However, a traffic signal agent that selfishly learns to reduce its own queue size may introduce a higher flow of vehicles arriving at neighboring intersections, thereby affecting the rewards of other agents and producing non-stationarity. We believe that in more complex scenarios this effect would be more clearly visible.

Furthermore, we found that traditional tabular Independent Q-learning presented a good performance in our scenario if we do not take into account the non-stationarity impacts. Therefore, in this particular simulation it was not necessary to use more sophisticated methods such as algorithms based on value-function approximation; for instance, deep neural networks. These methods could help in dealing with larger-scale simulations that could require dealing with higher dimensional states. However, we emphasize the fact that even though they could help with higher dimensional states, they would also be affected by the presence of non-stationarity, just like standard tabular methods are. This happens because just like standard tabular Q-learning, deep RL methods do not explicitly model the possible sources of non-stationarity, and therefore would suffer in terms of learning performance whenever changes in state transition function occur.

Related work

Reinforcement learning has been previously used with success to provide solutions to traffic signal control. Surveys on the area (Bazzan, 2009; Yau et al., 2017; Wei et al., 2019) have discussed fundamental aspects of reinforcement learning for traffic signal control, such as state definitions, reward functions and algorithms classifications. Many works have addressed multiagent RL (Arguello Calvo & Dusparic, 2018; Mannion, Duggan & Howley, 2016; El-Tantawy, Abdulhai & Abdelgawad, 2013) and deep RL (Van der Pol, 2016; Liang et al., 2018; Liu et al., 2017) methods in this context. In spite of non-stationarity being frequently mentioned as a complex challenge in traffic domains, we evidenced a lack of works quantifying its impact and relating it to its many causes and effects.

In Table 1 we compare relevant related works that have addressed non-stationary in the form of partial observability, change in vehicle flow distribution and/or multiagent scenarios. In (Da Silva et al., 2006), Da Silva et. al explored non-stationarity in traffic signal control under different traffic patterns. They proposed the RL-CD method to create partial models of the environment—each one responsible for dealing with one kind of context. However, they used a simple model of the states and actions available to each traffic signal agent: state was defined as the occupation of each incoming link and discretized into 3 bins; actions consisted of selecting one of three fixed and previously-designed signal plans. In (Oliveira et al., 2006), Oliveira et al. extend the work in (Da Silva et al., 2006) to address the non-stationarity caused by the random behavior of drivers in what regards the operational task of driving (e.g. deceleration probability), but the aforementioned simple model of the states and actions was not altered. In (Balaji, German & Srinivasan, 2010), Balaji et al. analyze the performance of tabular Q-learning in a large multiagent scenario. Their state state space, however, was significantly discretized and constituted of only 9 possible states. In (Liu et al., 2017), Liu et al. proposed a variant of independent deep Q-learning to coordinate four traffic signals. However, no information about vehicle distribution or insertion rates was mentioned or analyzed. A comparison between different state representations using the A3C algorithm was made in (Genders & Razavi, 2018); however, that paper did not study the capability of agents to adapt to different traffic flow distributions. In (Zhang et al., 2018) state observability was analyzed in a vehicle-to-infrastructure (V2I) scenario, where the traffic signal agent detects approaching vehicles with Dedicated Short Range Communications (DSRC) technology under different rates. In (Horsuwan & Aswakul, 2019) a scenario with partially observable state (only occupancy sensors available) was studied, however no comparisons with different state definitions or sensors were made. In (Chu et al., 2019), Chu et al. introduced Multiagent A2C in scenarios where different vehicle flows distributed in the network changed their insertion rates independently. On the other hand, they only used a state definition which gives sufficient information about the traffic intersection. Finally, in (Padakandla, Prabuchandran & Bhatnagar, 2019), Padakandla et al. introduce Context-QL, a method similar to RL-CD that uses a change-point detection metric to capture context changes. They also explored non-stationarity caused by different traffic flows, but they did not consider the impact of the state definition used (with low discretization and only one sensor) in their results. To the extent of our knowledge, this is the first work to analyze how different levels of partial observability affect traffic signal agents under non-stationary environments where traffic flows change not only in vehicle insertion rate, but also in vehicle insertion distribution between phases.

Table 1 Related work.

Study	Scenario	Method	State observability	Flow non-stationarity	
Da Silva et al. (2006) and Oliveira et al. (2006)	3 × 3 grid network	RL-CD (model-based)	Occupation discretized in 3 bins (no comparison made)	Two unbalanced flows	
Balaji, German & Srinivasan (2010)	Central Business District area in Singapore	Q-learning (model-free)	Queue and flow (9 possible states only)	Morning and afternoon peaks	
Liu et al. (2017)	2 × 2 grid network	CDRL (model-free)	Position, speed and neighbour intersection (no comparison made)	Not mentioned	
Genders & Razavi (2018)	Isolated intersection	A3C (model-free)	Three different definitions (different resolutions compared)	Variable flow rate equally distributed between phases	
Zhang et al. (2018)	Multiple network topologies	DQN (model-free)	Different car detection rates compared	Variable flow rate equally distributed between phases	
Horsuwan & Aswakul (2019)	Isolated intersection on Sathorn Road	Ape-X (model-free)	Mean occupancy (no comparison made)	Fixed flow	
Chu et al. (2019)	5 × 5 grid network	Multiagent A2C (model-free)	Delay and number of vehicles (no comparison made)	Time variant major and minor traffic flow groups	
Padakandla, Prabuchandran & Bhatnagar (2019)	Isolated intersection	Context QL (applicable for model-free and model-based)	Queue discretized in 3 bins (no comparison made)	High and low volume	
Ours	4 × 4 grid network	Independent Q-learning (model-free)	Queue and density (lack of sensors and different resolutions compared)	Two unbalanced flows	

Conclusion

Non-stationarity is an important challenge when applying RL to real-world problems in general, and to traffic signal control in particular. In this paper, we studied and quantified the impact of different causes of non-stationarity in a learning agents performance. Specifically, we studied the problem of non-stationarity in multiagent traffic signal control, where non-stationarity resulted from explicit changes in traffic patterns and from reduced state observability. This type of analysis complements those made in existing works related to non-stationarity in RL; these typically propose computational mechanisms to learn under changing environments, but usually do not systematically study the specific causes and impacts that the different sources of non-stationary may have on learning performance.

We have shown that independent Q-Learning agents can re-adapt their policies to traffic pattern context changes. Furthermore, we have shown that the agents state definition and their scope of observations strongly influence the agents re-adaptation capabilities. While agents with more extensive state observability do not undergo performance drops when dynamics change to previously-experienced contexts, agents operating under a partially observable version of the state often have to relearn policies. Hence, we have evidenced how a better understanding of the reasons and effects of non-stationarity may aid in the development of RL agents. In particular, our results empirically suggest that effort in designing better sensors and state features may have a greater impact on learning performance than efforts in designing more sophisticated learning algorithms.

For future work, traffic scenarios that include other causes for non-stationarity can be explored. For instance, unexpected events such as traffic accidents may cause drastic changes to the dynamics of an intersection, as they introduce local queues. In addition, we propose studying how well our findings generalize to settings involving arterial roads (which have greater volume of vehicles) and intersections with different numbers of traffic phases.

Additional Information and Declarations

Competing Interests

Author Contributions

Data Availability

1 A traffic phase assigns green, yellow or red light to each traffic movement. A green traffic phase is a phase which assigns green to at least one traffic movement.

The authors declare that they have no competing interests.

Lucas N. Alegre conceived and designed the experiments, performed the experiments, analyzed the data, performed the computation work, prepared figures and/or tables, authored or reviewed drafts of the paper, and approved the final draft.

Ana L.C. Bazzan conceived and designed the experiments, analyzed the data, authored or reviewed drafts of the paper, and approved the final draft.

Bruno C. da Silva conceived and designed the experiments, analyzed the data, authored or reviewed drafts of the paper, and approved the final draft.

The following information was supplied regarding data availability:

The implementation of the algorithms and SUMO traffic scenario is available at GitHub: https://github.com/LucasAlegre/sumo-rl.

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
