# Peer review of "Quantifying the impact of non-stationarity in reinforcement learning-based traffic signal control"

_PeerJ Computer Science, doi:10.7717/peerj-cs.575_

## Round 0.1 · original submission · Major Revisions

Based on the review reports, the paper needs an essential revision.

Reviewer 1 ·

Basic reporting

The manuscript is written and organised well. Apart from a few typos and grammar issues, readability is very good. Please correct situations such as:

"We them analyze..." -> "We then analyze..."?
"the impact that each one has in the learning process..." -> "on the learning process"?
"policy π^∗The rewards" -> a full stop and space are perhaps missing.

Authors also provide sufficient background underlying the preliminaries of their approach and discuss the related work to emphasise on their contributions. References seem to be appropriate.

Figures/graphics/plots are also readable and easily understandable. And results are representative of and in accordance with the authors' methodological approach and discussion.

Experimental design

Authors are interested in analysing the effect of causes of nonstationarity in their model-free RL approach to adaptive traffic signal control. Causes of nonstationarity, as listed by the authors in their study, include traffic flow pattern and the partial-observability ability of traffic signal agents. With this goal in mind, authors present different simulation setups to test their RL traffic signal agents in a grid-like traffic network modelled under the SUMO microscopic simulator.

In this perspective, some concepts might be better discussed, though. Whereas traffic flow in urban settings is potentially nonstationary, it is also true that some patterns (although stochastic) are easily observed, such as the daily 24-hour flow profile, in which off-peak and peak hours are identified. Stochasticity not necessarily implies nonstationarity, as some systems may well work under dynamic stability. Certainly, such dynamic stability may well be disturbed by unexpected events, such as accidents, or even by priority vehicles like ambulances demanding other vehicles to give the right of way. From what this reviewer could understand, it seems that the "nonstationary traffic flow" is rather yielded by a random OD demand imposed by authors in their different simulation scenarios. On the other hand, limitations in agents' observability capacity do not seem to cause itself nonstationary regimes upon the environment as might do agents' actions. Rather than a cause, it seems to be related to how sometimes such limitation might well influence rationality even in (dynamic) stationary environments making them seem rather nonstationary to the sensor-deficient agents, affecting this way the soundness of their decision making.

Validity of the findings

Results are promising (although rather intuitive) and are discussed quite well. Whether being able to observe (sense) "densities", perhaps this might also be influenced by how often vehicles' speed drops below 0.1 m/s on a link and by the distance to be traversed. The best-case scenario is when a vehicle traverses a link on free-flow speed and does not need to stop at the junction; performance degrades when flow approaches saturation, affecting densities. Isn't this behaviour of the link performance implicitly embedded in the queue signal perceived by the agents?

Also, authors have selected Total Waiting Time as the main metric to analyse. Perhaps considering total/average travel time for the whole network would allow for evaluating the system's performance as a whole (collectively). That could be an interesting analysis too.

Additional comments

Generally, the work is quite relevant and makes a good appreciation of related efforts to emphasise the authors' contributions. Some aspects, especially conceptual ones, could improve benefiting from a better discussion and deeper reflection.

Reviewer 2 ·

Basic reporting

no comment

Experimental design

no comment

Validity of the findings

no comment

Additional comments

This paper applies a multiagent independent Q-learning method to study the traffic signal control, especially for non-stationarity and partial observability. It is an interesting job but it is not clear for the key of end condition in non-stationarity and partial observability. As is known the reinforcement learning is a class of iterative learning of episodes based on rewards and terminated at a training convergence. Authors declare “This means that original convergence properties for single-agent algorithms no longer hold due to the fact that the best policy for an agent changes as other agents’ policies change” (line 166). What is a performance guarantee mechanism instead of convergence during the non-stationarity iterative process? And how to provide? I notice the experiment takes use of times steps as end condition. Obviously it is not universal.

---

## Round 0.2 · Major Revisions

The reviewer still has concerns. As a result, a further revision is recommended.

Reviewer 2 ·

Basic reporting

no comment

Experimental design

no comment

Validity of the findings

no comment

Additional comments

I am sorry for my previous unclear statement. The RL is used to find an optimal action that matches the current states and the stationarity or non-stationarity environment in a long run. The RL is trained by episodes to pursue rewards (actions value), which is classified as an online mode and an offline mode. For online the reward of each episode can be obtained from the Q table (descrete system) or the critic network (continuous system) based on observations after taking an action. But no one knows the effect of this action in training process, which will lead to a fatal fault (traffic confusion). A feasible way is offline training. But it needs a model of actions on the environment to get the action value in all kinds of traffic cases. This model (missing in this paper) is indispensable for adjusting the Q table or the weights of critic network. Both training processes will suffer from an early fluctuation to the end stability (this is called to be convergent). For a stationarity, it will spend a very long time to get the optimal action. For a non-stationarity, maybe it will keep changing and cannot get optimal action. So I ask for how to provide a performance guarantee mechanism. By the way the agent will get the optimization (be convergent) if given sufficient time seems not to meet the reality.

---

## Round 0.3 · accepted · Accept

The reviewer is happy with the revision. As a result, the paper is ready to be accepted.

Reviewer 2 ·

Basic reporting

no comment

Experimental design

no comment

Validity of the findings

no comment

Additional comments

The concerns are responded. Thanks.